# Vehicle-in-the-Loop in Global Coordinates for Advanced Driver Assistance System

**Changwoo Park [1] , Seunghwan Chung [2] and Hyeongcheol Lee [3],***

1   Department of Electrical Engineering, Hanyang University, Seoul 04763, Korea; changwoo@hanyang.ac.kr
2   Autonomous Driving Simulation Research Lab., Hyundai Motor Company, 150 Hyundaiyeonguso-ro, Namyang-eup, Hwaseong-si 18280, Korea; chungsh@hyundai.com
3   Department of Electrical and Biomedical Engineering, Hanyang University, Seoul 04763, Korea
*   Correspondence: hclee@hanyang.ac.kr; Tel.: +82-2-2220-1685

**Abstract:** Most vehicle controllers are developed and verified with V-model. There are several traditional methods in the automotive industry called "X-in-the-Loop (XIL)". However, the validation of advanced driver assistance system (ADAS) controllers is more complicated and needs more environmental resources because the controller interacts with the external environment of the vehicle. Vehicle-in-the-Loop (VIL) is a recently being developed approach for simulating ADAS vehicles that ensures the safety of critical test scenarios in real-world testing using virtual environments. This new test method needs both properties of traditional computer simulations and real-world vehicle tests. This paper presents a Vehicle-in-the-Loop topology for execution in global Coordinates system. Also, it has a modular structure with four parts: synchronization module, virtual environment, sensor emulator and visualizer, so each part can be developed and modified separately in combination with other parts. This structure of VIL is expected to save maintenance time and cost. This paper shows its acceptability by testing ADAS on both a real and the VIL system.

**Keywords:** advanced driver assistance system; autonomous driving; model-based development; vehicle in the loop; vehicle validation

---

## 1. Introduction

Recently, Advanced Driver Assistance Systems (ADAS) and Autonomous Driving (AD) have been actively developed to increase the convenience and safety of the driver and passengers. The systems that are closely linked to the safety of the driver and the occupant will need to be developed more cautiously [1,2]. There are effective development tools for each development stage of the vehicle controller. Model-in-the-Loop (MIL) is very useful for quickly verifying and developing algorithms by implementing a controller as a computer simulation model. Software-in-the-Loop (SIL) is usually verified for pseudo-code in a form that can be embedded in an actual automotive controller. Both tests can be simulated faster than real-time, so many scenarios can be verified in a short time. Next, Hardware-in-the-Loop (HIL) validates the actual ECU design and operating characteristics at the hardware level in a laboratory environment. Finally, the verification of the completed system is performed at the actual vehicle stage. In the actual vehicle phase, the most reliable verification is possible, including vehicle dynamic characteristics. For each step, the time required for verification and reliability are regarded as a trade-off relationship, as in Figure 1 [2–6].

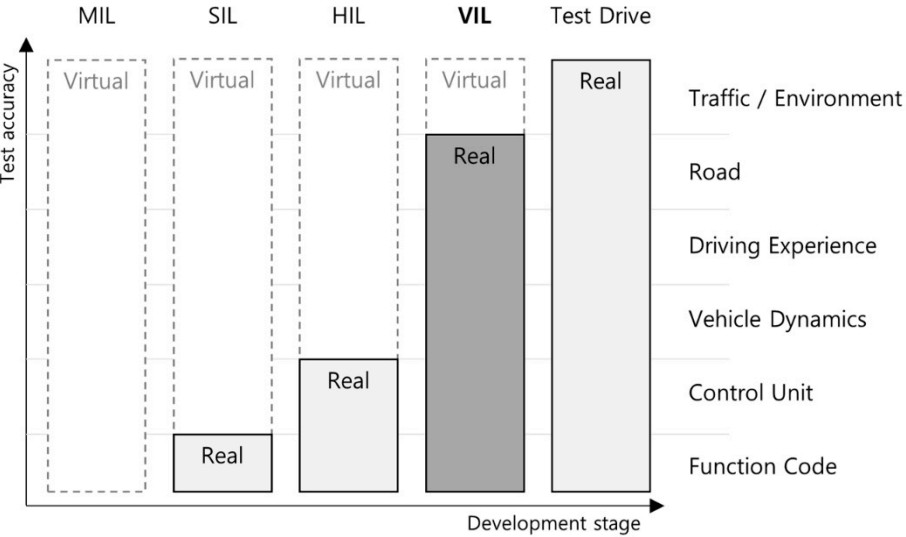

**Figure 1.** MIL vs. SIL vs. HIL vs. VIL vs. Test Drive.

Normally, stand-alone vehicle testing after the MIL–SIL–HIL process is acceptable for conventional vehicle controllers—headlamps, wipers and smart keys in a body control system, and Anti-lock Brake System (ABS), Electronic Stabilization Control (ESC) and Electric Power Steering (MDPS) in a chassis control system.

In the case of ADAS, however, external environments such as surrounding vehicles, pedestrians, and lanes should be accompanied as a factor in the verification. To overcome these limitations, new approaches have been proposed to validate ADAS. For driver acceptance in the ADAS vehicle, Driver-in-the-Loop (DIL) and Human-and-Hardware-In-the-Loop (H2IL) are suggested as cyber-physical systems [7,8]. DIL has conducted a fixed-base driving simulator with TNO PreScan software and is used for testing the human–automation interaction design of the lane-keeping assistance system (LKAS) and adaptive cruise control (ACC). H2IL has components to generate, operate and communicate with the virtual environment as a driving simulator. It is designed to analyze driving performance and safety affected by V2X communication support. The Vehicle-Hardware-in-the-Loop (VEHIL) is operated on the chassis dynamometer and simulates the surrounding environment with a moving base and a front display device each for a radar and vision sensor [9–11]. Additionally, the Vehicle-Traffic-Hardware-in-the-Loop (VTHIL) robots validated a radar sensor using a moving base and a commercial vision sensor using a front display device [12]. These can verify the sensor effectiveness, but there is a limit to reflect the own vehicle dynamic characteristics. In addition, the cost of construction and maintenance is high, and it is difficult to reuse and modify the built system when verification is not in accordance with the initial system. Alternatively, building a proving ground for ADAS is suitable for the recognition and judgment of the sensor and the dynamic characteristics of the own, and it can be verified very similarly to the real environment. The proving ground needs a test road dedicated to ADAS and to simulate objects with a robot or a dummy [13,14]. However, the test road construction and the individual test costs are very high. There is also a high risk of collision by surrounding objects during testing.

In order to solve these problems, the Vehicle-in-the-Loop (VIL) technique is emerging to construct and utilize a virtual environment. VIL is a fusion environment of a real testing vehicle in the real-world and a virtual environment, as shown in Figure 2a. It can reflect vehicle dynamics at the same level as the real-world and save the cost of constructing an external environment for system verification. At the same time, the risk of collision with external objects that may occur during testing can be eliminated. Thomas Bock proposed a hardware configuration that links augmented reality to existing vehicle simulators [15,16]. Laschinsky has shown that it can be applied in the development of active safety lights to conduct tests in the daytime [17]. VIL is used for driver reaction of critical situations [18–21].

Additionally, Miquet and Schwab suggested VIL with a head-mounted display (HMD) device for the parking assistance system. [22,23]. Griggs validated a speed advisory system via VIL [6,24]. Fayazi developed a VIL program for simulation of intersection controller [25,26]. Tettamanti and Rastogi showed a VIL simulation environment that validates ADAS controllers with virtually preceding vehicles [27,28]. Horváth used VIL as a traffic controller simulator and Che suggested a VIL with the control center and connectivity [29,30].

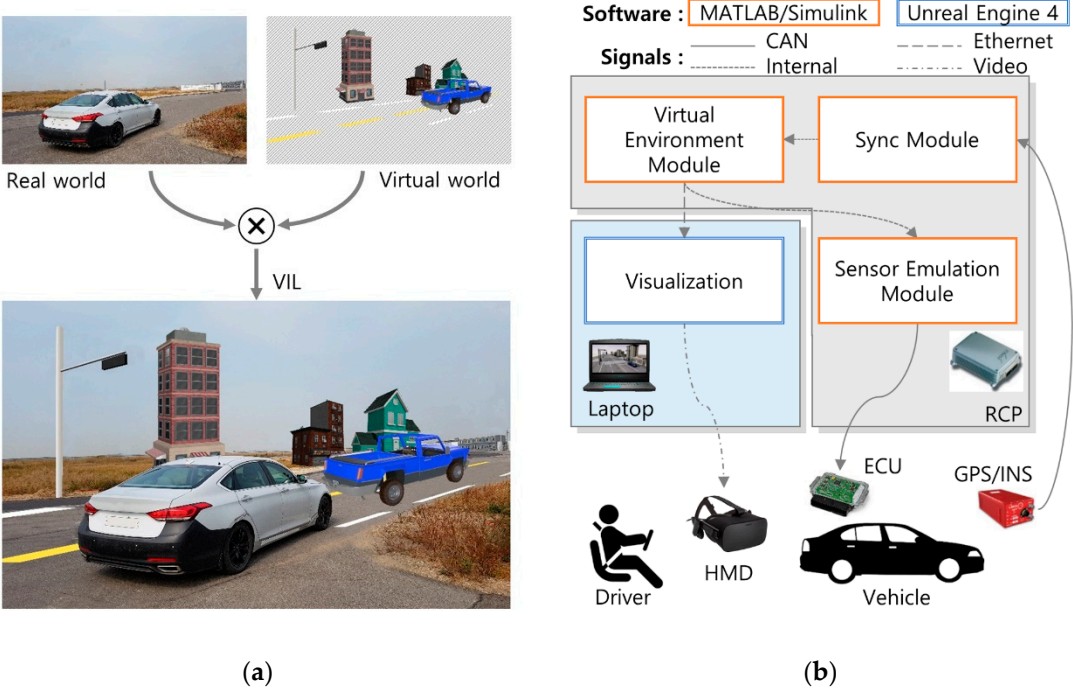

(**a**)                                           (**b**)

**Figure 2.** (**a**) Concept of VIL; (**b**) Configuration of VIL.

Previous studies have suggested the composition and application of VIL. However, the working algorithm of VIL for ADAS simulation was not published and only showed the environment in which the study was conducted. The studies in [17–21,31] use VIRES's Virtual Test Drive (VTD) without revealing the algorithm. [22,23] and [28] also constructed their VIL system with commercial simulation tools, CarMaker and CarSim. [6,24,27,29] verified their target system based on SUMO as an opensource tool, but this tool does not cover ADAS with buildings, pedestrians and road lines as targets. VIL in the research [25,26] seems to have its own tools, however, the VIL is programmed for intersection controller simulation and provides only limited information, not available to the ADAS controller.

There is no research suggesting an effective development algorithm for VIL. In this paper, we propose a VIL topology that can be developed and modified independently by separating each module. Although the ADAS controller is considered a vehicle-based local coordinates system, the VIL has been developed as a global coordinates system for connectivity with virtual environments. In Section 2, we present the basic structure of VIL and the essential parameters of each module. In Section 3, we compare the method of integrating each module and the similarity test between the actual ADAS vehicle and VIL to confirm whether the proposed system meets the VIL purpose. Finally, Section 4 describes the module variability of the proposed VIL system.

## 2. Vehicle in the Loop

The VIL topology in this paper consists of four modules, depending on their characteristics, and defines the elements required for each module to work organically, as shown in Figure 2b. One of the components is a virtual environment module that contains information of the surrounding vehicles, pedestrians, and road lines. VIL also requires a synchronization module for synchronizing



the real environment with the virtual driving environment and a sensor emulation module for transmitting virtual environment information to the ADAS controller in the real world. Finally, the driver visualization module provides the driver with a visual virtual driving environment. Each module is described in Section 2.1 to Section 2.4. Individual modules can be replaced with commercial tools or used flexibly to meet the needs of the researcher.

## 2.1. Virtual Environment

The virtual environment module is the configuration of elements for VIL testing. For example, the advanced emergency brake (AEB) and adaptive cruise control (ACC) controllers relate to surrounding vehicles and pedestrians. For the lane keeping-assist system (LKAS) controllers, the road lines are closely related. In addition, high-level autonomous vehicles are subject to additional elements such as traffic lights, signs and buildings. For optimal classification, we divide these objects into time-invariant (TI) and time-varying (TV) objects. Time-varying objects are further divided into time-varying-pose (TVP) and time-varying-attribute (TVA), depending on what information changes. Table 1 shows each object type and the stored data. Below are examples of each object:

- TI Object: buildings, signs, roads (lines);
- TVP: vehicles, bicycles, pedestrians;
- TVA: traffic lights.

**Table 1.** Data of object type.

| Object | Time Invariant Data | Time Varying Data |
|---|---|---|
| T.I. Object | Attribute, Pose | - |
| T.V. Pose-Object | Attribute | Pose |
| T.V. Attribute-Object | Pose | Attribute |

All types of objects have object attribute and pose data. Attribute data have the type, nodes of shape, and additional information of the object. It can be configured differently depending on the characteristics of each object. The pose data contain position and orientation information such as x, y, and yaw. Additionally, z, pitch, and roll can be included, but in this paper, the virtual environment is designed based on the 2D plane. The variables in Tables 2–5 are superscripted with object type and subscripted with variable order.

**Table 2.** Example of a building as the time-invariant object.

| Type | Data |
|---|---|
| Attribute | Building$/\left(nx_1^B, ny_1^B\right), \left(nx_2^B, ny_2^B\right), \ldots, \left(nx_k^B, ny_k^B\right)$ |
| Pose | $x_0^B, y_0^B, \psi_0^B$ |

**Table 3.** Example of a road line as the time-invariant object.

| Type | Data |
|---|---|
| Attribute | White-Dot-Line$/\left(nx_1^L, ny_1^L, \theta_1^L, C_1^L, D_1^L\right), \ldots, \left(nx_k^L, ny_k^L, \theta_k^L, C_k^L, D_k^L\right)$ |
| Pose | $x_0^L, y_0^L, \psi_0^L$ |

**Table 4.** Example of a vehicle as the time-varying pose object.

| Type | Data | | | |
|---|---|---|---|---|
| Attribute | passenger car$/\left(nx_1^V, ny_1^V\right), \left(nx_2^V, ny_2^V\right), \ldots, \left(nx_k^V, ny_k^V\right)$ | | | |
| Pose | $x_0^L(0), y_0^L(0), \psi_0^L(0)$ | $x_0^L(1), y_0^L(1), \psi_0^L(1)$ | $\ldots$ | $x_0^L(n), y_0^L(n), \psi_0^L(n)$ |
| Time | $t(0)$ | $t(1)$ | $\ldots$ | $t(n)$ |

**Table 5.** Example of a traffic light as the time-varying attribute object.

| Type | Data | | | |
|---|---|---|---|---|
| Pose | $x_0^T, y_0^T, \psi_0^T$ | | | |
| Attribute | $color(0)$ | $color(1)$ | ... | $color(n)$ |
| Time | $t(0)$ | $t(1)$ | ... | $t(n)$ |

Table 2 indicates that this object is a building. It also has sets of node distances from the object center and the pose (position and orientation) of the object. The shape nodes, where k $(x_k^B, y_k^B)$ of the building, can be represented in the following general equation. The object shape node $(x_k, y_k)$ can be obtained by using (1), where the center position and the yaw angle of the object are $x_0$, $y_0$ and $\psi_0$, and the set of node distance is $(nx_k, ny_k)$, as shown in Figure 3.

$$\begin{bmatrix} x_k \\ y_k \end{bmatrix} = \begin{bmatrix} x_0 \\ y_0 \end{bmatrix} + \begin{bmatrix} cos(-\psi_0) & -sin(-\psi_0) \\ sin(-\psi_0) & cos(-\psi_0) \end{bmatrix} \begin{bmatrix} nx_k \\ ny_k \end{bmatrix} \tag{1}$$

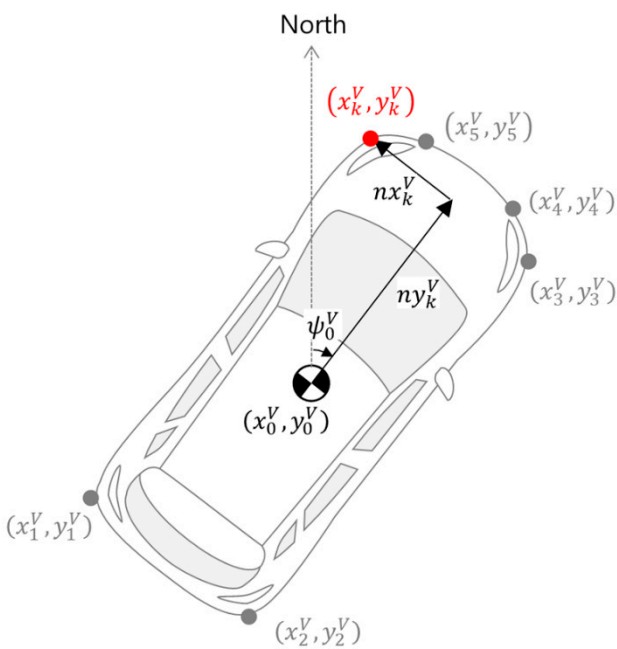

**Figure 3.** Node coordinates.

The road line object in Table 3 contains not only line style and node distances but also the angles $(\theta_k^L)$ from the original angle, curvature $(C_k^L)$ and curvature derivative $(D_k^L)$ of each point k of line. This object can be represented as a continuous line.

The time-varying pose object in Table 4 has its attribute information as static, and the pose data as time series. In many cases, TVP objects are vehicles driving around. For simple test scenarios, these can use predefined scenarios. However, the interaction of each vehicle, including the ego vehicle, may be required for a more advanced test environment. To implement this advanced environment, there needs to be a vehicle and driver model in a closed-loop system. In our case, the models can be designed simple enough for the specificity of scenario creation and optimization of computation time. For the simple vehicle motion, the bicycle kinematic model and the slip angle $\beta$ are below.

$$\begin{bmatrix} \dot{x} \\ \dot{y} \\ \dot{\psi} \end{bmatrix} = \begin{bmatrix} cos(\psi + \beta) & 0 \\ sin(\psi + \beta) & 0 \\ 0 & \frac{vcos(\beta)tan(\delta)}{L} \end{bmatrix} \begin{bmatrix} v \\ \delta \end{bmatrix} \tag{2}$$

$$\beta = \tan^{-1}\left(\frac{l_r tan(\beta)}{L}\right) \tag{3}$$

$v$ and $\delta$ are the longitudinal velocity and the steering wheel angle. Additionally, $l_r$ and $L$ are the distance from the rear wheel to the center of gravity (CG) and the wheelbase. To generate the velocity and steering wheel angle, a driver model is still needed. It is acceptable to choose any driver model by previous studies. One of the representative driver models for traffic is Gipps car-following model and lane-changing decision model [32,33]. The car-following model generates speed in response to situations caused by other vehicles. The lane-changing model considers physical possibilities, necessary and desirable to change lane. If Gipps lane-changing model determines the lane change, then it can be controlled using a simple sinusoid or polynomial input.

Lastly, there is a time-varying attribute object that changed its attribute like traffic signal light sequence as in Table 5. Each object's information in the virtual environment is updated every step time so that the dynamics of the objects can be used in another module.

## 2.2. Sync

This synchronization module allows the movement of a real vehicle in a virtual environment. For this purpose, the Global Positioning System (GPS) and Inertial Navigation System (INS) devices are mounted onto the test vehicle. The position measured by the GPS device is usually in the form of a spherical coordinate system, which is a coordinate system of longitude and latitude. However, the conversion to the Cartesian coordinate system is useful for intuitive system development, user acceptance and improving system connectivity. The algorithm is included in the synchronization module. In this paper, WGS84 (World Geodetic System 1984) coordinates obtained from GPS are converted to Transverse Mercator (TM) using the following method.

$$x^{EGO} = (R_N + h)(\phi - \phi_0) + \Delta X \tag{4}$$

$$y^{EGO} = (R_E + h)(\lambda - \lambda_0) + \Delta Y \tag{5}$$

$\{\phi_0, \lambda_0\}$ is the central origin constant for longitude and latitude, and $\phi$, $\lambda$, $h$ are the longitude, latitude, and altitude of the vehicle's position as measured by GPS. $\{\Delta X, \Delta Y\}$ is the origin added values, (e.g., 2,000,000 and 6,000,000 set by the National Geographic Information Service in Korea). However, in this study, it is aimed to test a vehicle which is carried out in a narrow area, so that the added values of the origin can be modified to match the test location. The radius of curvature in the north/east direction is calculated using the major axis ($R_{major}$) and eccentricity (e) of the earth ellipse.

$$R_N = \frac{R_{major}(1-e)^2}{\left(1 - e^2 sin^2\phi\right)^{3/2}} \tag{6}$$

$$R_E = \frac{R_{major}}{\left(1 - e^2 sin^2\phi\right)^{1/2}} \tag{7}$$

The relationship between eccentricity (e) and flatness ($f$) is as follows. The major axis ($R_{major}$) and minor axis ($R_{minor}$) of the earth are used for the calculation.

$$e = \sqrt{1 - (1-f)^2} = \sqrt{1 - \left(1 - \frac{R_{major} - R_{minor}}{R_{major}}\right)^2} \tag{8}$$

The yaw of the vehicle ($\psi^{EGO}$) can directly use the data measured by the INS device.

## 2.3. Sensor Emulation

ADAS controllers receive information on objects that have been recognized and judged from the radar or vision sensors via Control Area Network (CAN). However, in the VIL environment, there are no real objects that real sensors can recognize. Thus, the sensor emulation module uses the information of the virtual object and the location of the actual synchronized test vehicle to process it. For example, radar sensor emulators are used to calculate relative distance, speed, and azimuth values for synchronized test vehicles and vehicles or pedestrians. The vision sensor emulator processes left and right road lines, and these values are sent to the real controller. The radar sensor emulator generates the signal processed through the shape points $\{x_k^V, y_k^V\}$ of the virtual object and the sensor pose $\{x_S^{EGO}, y_S^{EGO}, \psi_S^{EGO}\}$ on the real test vehicle. These points and this pose can be obtained by using (1). The relative distances and azimuth angles of the objects are determined by Euclidean geometry using the $x_k^V, y_k^V, x_S^{EGO}, y_S^{EGO}$, and $\psi_S^{EGO}$. We can define nodes positions $N_1$ and $N_2$ on the object, sensor position $S$, and the position $H$ on the object located at the shortest distance between the sensor and the object. Additionally, $T$ is a temporary point to obtain $H$. These are shown in Figure 4a.

$$N_1 = \begin{bmatrix} x_1^V \\ y_1^V \end{bmatrix}, \; N_2 = \begin{bmatrix} x_2^V \\ y_2^V \end{bmatrix}, \; S = \begin{bmatrix} x_s^{EGO} \\ y_s^{EGO} \end{bmatrix}, \; H = \begin{bmatrix} x_h^V \\ x_h^V \end{bmatrix}, \; T = \begin{bmatrix} x_t^N \\ y_t^N \end{bmatrix} \tag{9}$$

The shortest distance between the sensor position and a line that includes two nodes can be obtained through the norm as shown in (10), and the position is calculated by (11).

$$\| \overrightarrow{SH} \| = \frac{\overrightarrow{N_1N_2}}{\| \overrightarrow{N_1N_2} \|} \cdot \overrightarrow{N_1S} \tag{10}$$

$$T = N_1 + \| \overrightarrow{ST} \| \times \frac{\overrightarrow{N_1N_2}}{\| \overrightarrow{N_1N_2} \|} \tag{11}$$

The position $H$ must exist between the two nodes $N_1$ and $N_2$. Therefore, it can be selected as follows.

$$N_k = \begin{cases} N_1, & \| \overrightarrow{N_1S} \| < \| \overrightarrow{N_2S} \| \\ N_2, & else \end{cases} \tag{12}$$

$$H = \begin{cases} T, & x_t^N \in \left[ x_1^V, x_2^V \right] \wedge y_t^N \in \left[ y_1^V, y_2^V \right] \\ N_k, & else \end{cases} \tag{13}$$

Finally, the shortest distance and the recognized azimuth angle of the object is:

$$d_{SH} = \| \overrightarrow{SH} \| \tag{14}$$

$$\theta_{SH} = tan^{-1}\left( \frac{x_h^V - x_s^{EGO}}{y_h^V - y_s^{EGO}} \right) - \theta_s^{EGO} \tag{15}$$

The $d_{SH}$ and $\theta_{SH}$ are idle results from the geometric model and cannot represent perception in real environment. For the more realistic, it can be added that some noise caused by component tolerances, temperature drifts and quantization, as shown in (16):

$$\begin{bmatrix} d_{noisy} \\ \theta_{noisy} \end{bmatrix} = \begin{bmatrix} d_{SH} \\ \theta_{SH} \end{bmatrix} + \begin{bmatrix} N(0, \sigma_d^2) & N(0, \sigma_\theta^2) \end{bmatrix}^T \tag{16}$$

$N\left(0, \sigma_d^2\right)$ and $N\left(0, \sigma_\theta^2\right)$ represent random numbers with a mean of 0 and standard deviations of $\sigma_d$ and $\sigma_\theta$. The standard deviations are linearly dependent on the distance and angle from the sensor, and more detail is given in [34].

The vision sensor emulator transmits information from the virtual environment as images. In this paper, however, only line recognition is considered for a lateral control system. This information can be generally defined by the distance $A^L$, incidence angle $B^L$, the curvature of the lane $C^L$, and the rate of curvature changing $D^L$. The distances between the line and the ego vehicle and the incidence angle can be calculated using (17) and (18). The position $\left\{x_{FW}^{EGO}, y_{FW}^{EGO}\right\}$ is generally defined as the center axis of the front wheel in the ego vehicle to obtain the relative distance.

$$A^L = \sqrt{\left(x^L - x_{FW}^{EGO}\right)^2 + \left(y^L - y_{FW}^{EGO}\right)^2} \tag{17}$$

$$B^L = \psi^{EGO} - \psi^L \tag{18}$$

In addition, the curvature and curvature derivatives, which are line-specific information, can directly use the information defined in Table 3. These can be derived in advance through polynomial curve fitting, as shown in Figure 4b.

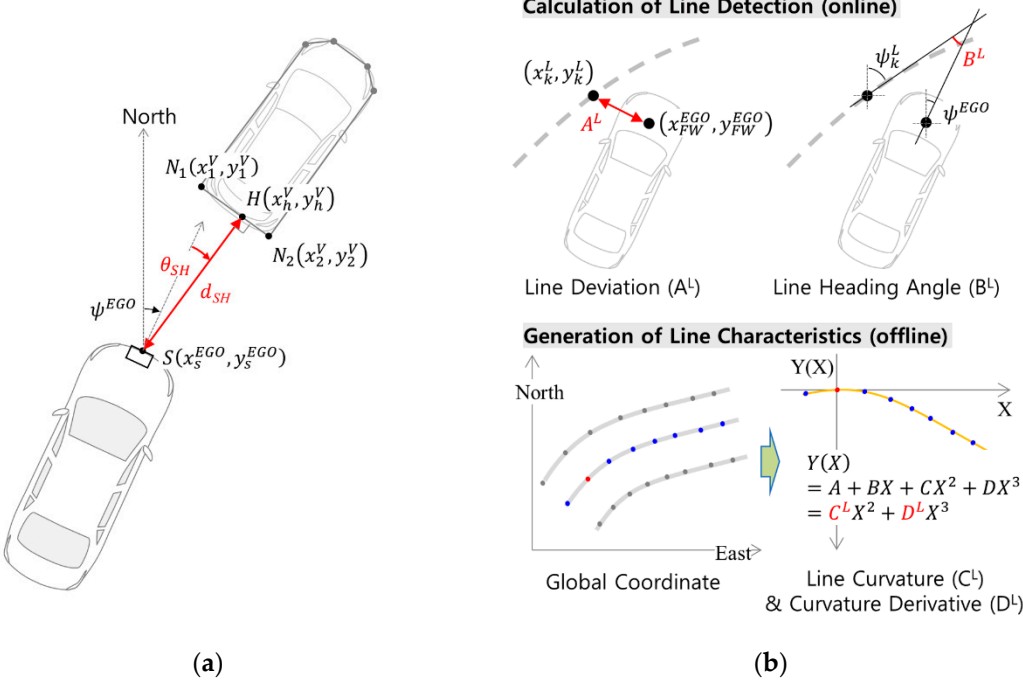

**(a)**　　　　　　　　　　　　　　　　　　　　　　**(b)**

**Figure 4.** (**a**) Calculation of radar detected object information; (**b**) Calculation of road line information by the vision sensor.

In the real world, the road line information is calculated with a certain section of the line that is ahead of the ego vehicle and there are environmental impact areas such as shadows, light, occlusions and standing water, as shown in Figure 5 [35,36]. When some of the interesting line points exist in the area, the perception rate of the vision sensor can be reduced. To extract the affected line points, some algorithms, called the point in polygon (PIP), are given in the previous studies. Two of the well-known ones are the ray casting algorithm and the winding number algorithm [37–40]. A set of perceptible $P_{out}$ that exists outside is extracted from the interesting points using the algorithm in (19).

$$P_{out} = PIP_{out}(P, A) \tag{19}$$

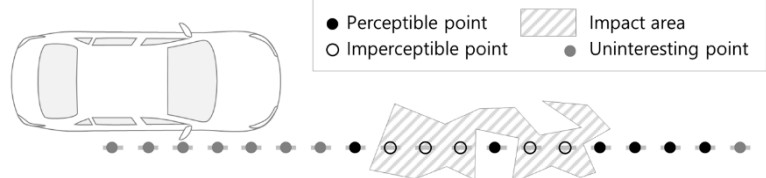

**Figure 5.** Perceptible and imperceptible points with impact area.

*P* and *A* represent each set of interesting line points and area node points. Detail of this algorithm is discussed in Appendix A. The derivable conditions for the road line information can be substituted with experimental values.

The recognition and judgment information generated by the radar and vision sensor emulation module can replace the real sensor for objects that do not actually exist.

*2.4. Visualization*

Real-time information of the virtual environment needs to be provided to the driver so that the convenience of test execution and the driving sensibility can be evaluated. This requires a module to visualize the virtual driving environment and this module provides a live view from the driver's perspective. The visualization module uses the attributes and poses of all objects, including the test vehicle. Recently, various tools for developing such a visualization program exist and can be easily implemented. An example of this is discussed in detail in Appendix B.

**3. Field Test**

Except for the visualization module, we implemented the VIL system based on MATLAB/Simulink. This program is a widely used tool for vehicle controller development and has a low barrier to entry. The 3D visualization module is programmed with Unreal Engine 4, developed by Epic Games, which allows the development of games efficiently. In recent years, it has been widely used outside of gaming, especially for visualization of computer simulation [41–43]. Unreal Engine 4 also has a programming language called Blueprint, similar to MATLAB/Simulink. Therefore, the same benefits exist with development accessibility.

The following hardware is equipped to configure for the VIL evaluation environment:

- GPS/INS: RT3100 by OxTS
- RCP: MicroAutoBoxII by dSPACE
- Laptop: ALIENWARE by Dell
- HMD: Oculus Rift by Oculus

VIL must ensure real-time stability. For this purpose, modules programmed with MATLAB/Simulink (virtual environment, synchronization, and sensor emulation) work on a Rapid Control Prototyping (RCP) device. RCP is highly integrated with MATLAB/Simulink, which helps to reduce development costs. It also uses high-resolution GPS/INS to synchronize the test vehicle to the virtual environment. To increase driver realism, we use high-performance gaming laptops and head-mounted display (HMD) devices. As can be seen in Figure 2b, each hardware device is connected in several ways. The pose information of the test vehicle, measured by GPS/INS, is transmitted to the RCP via CAN messages The RCP synchronizes the test vehicle with the virtual environment, emulates sensor data and sends the results to the real-world ADAS controller every 10 to 20 ms depending on the requirements. At the same time, the visualization program receives the pose information of all objects via ethernet every 10 ms. The data contain each position $x$ and $y$ in 0.01 m resolution with more than thousands of km coverage, and yaw angle $\psi$ in 0.01° resolution with 360° coverage. Scheme 1 shows the VIL hardware configuration and the VIL test scene.

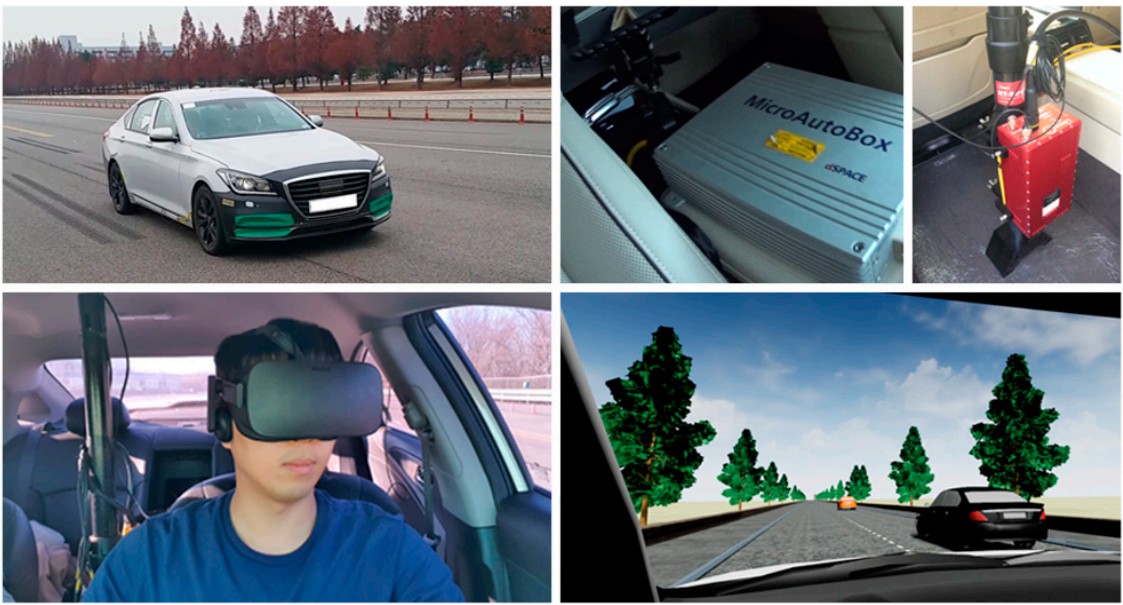

**Scheme 1.** The test vehicle, RCP, GPS/INS, HMD as VIL equipment and the testing scene.

In order to verify the proposed VIL in the global coordinates system, tests were performed on AEB and LKAS controllers. The Hyundai Genesis DH model equipped with the ADAS controllers was used as the test vehicle. First, the AEB test was repeated five times for each real and VIL environment in the same scenario. The scenario is selected as Car-to-Car Rear Stationary (CCR) according to the Euro (European New Car Assessment Programme) NCAP protocol. The ego vehicle travels at a speed of 10 m/s to the stationary target vehicle. As a result of the test, after the vehicle was stopped by AEB, the average distance to the target vehicle was measured. As Figure 6a shows, it is 2.46 m in the real target testing and 2.38 m in the testing with VIL. The difference in the results between the two environments is about 0.08 m, which is very reliable. In particular, it is shown that the maximum error occurs in 1.24 m and 0.63 m respectively in five tests performed in each test environment. If the sensor emulator is very ideal, the two maximum errors are considered due to the vehicle's dynamics uncertainty. In other words, the difference in results can be acceptable. The LKAS test was performed at a speed of about 20 m/s and a width of 3.65 m in a straight lane approaching the left line with a lateral speed of 0.4 m/s. At this time, the distances between the vehicle and the left and right lines measured by the real vision sensor and vision sensor emulator were compared. The distance between the left and right lines recognized in Figure 6b is very similar; the maximum distance error is 0.05 m where the ground truth is about 1.22 m. This is about a 4% error, which is lower than the error of the actual vision sensor, so it does not affect the whole system.

It is necessary to confirm driver heterogeneity according to the latency of the visualization module. However, it is difficult to verify directly from the structure of the HMD worn by the driver. As an alternative, we analyzed image changes on a laptop display connected to the HMD with a 60 FPS camera to delay data transfer. As a result, the difference in 1 frame is about 1/60 s. The driver who participated in the actual test did not recognize the delay.

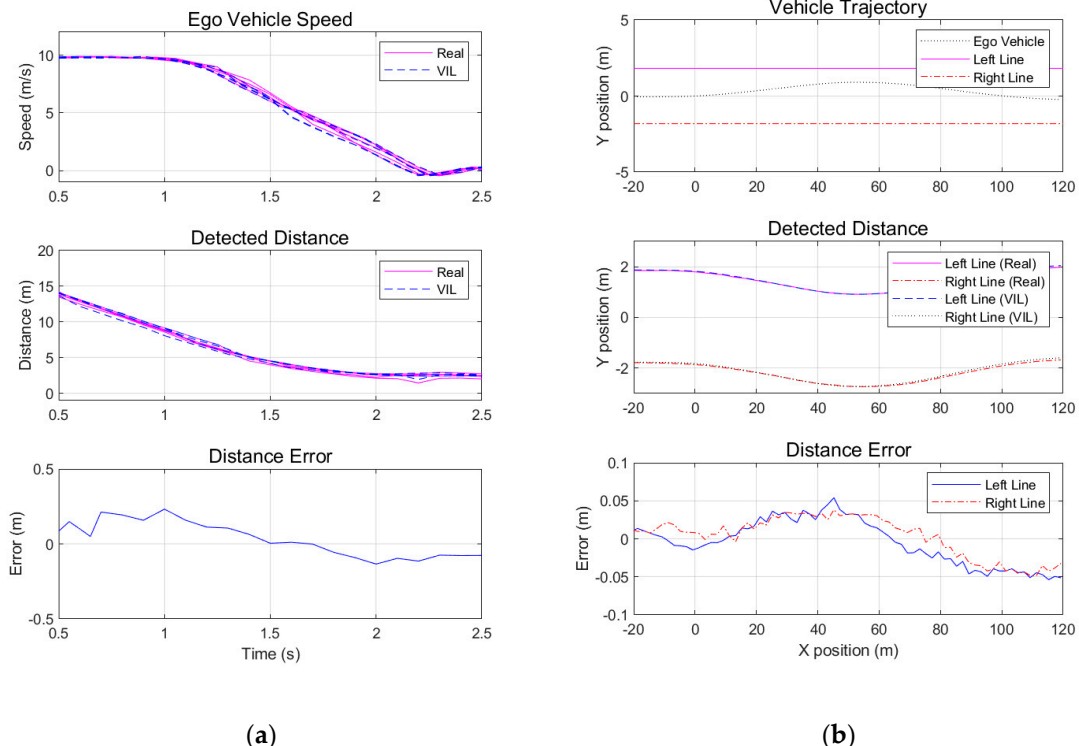

**Figure 6.** Testing result of (**a**) AEB with a radar sensor; (**b**) LKAS with a vision sensor.

## 4. Discussion

In this paper, we define the modular topology of the proposed VIL and show its connectivity with the virtual environment, synchronization, sensor emulation, and driver visualization modules. This module can be reused, even if certain modules are changed as needed. For example, to simulate the behavior of a virtual object, the method of real-time application of own vehicle models can be considered. In addition, sensor emulation modules can be complex depending on the physical characteristics of each sensor. It is also possible to replace the software model to a HIL system [2]. For driver visualization modules, they can be changed to another platform such as Direct X or Unity [28]. As mentioned above, the development and modification of each module can be carried out independently of the different modules, whereby the maintenance costs of the system are expected to be reduced.

In Section 3, we performed VIL tests on the AEB and LKAS scenarios and showed similarities in practice. However, due to the limitations of the research environment, only some ADAS scenarios are considered. In the future, it is necessary to acquire various advanced scenarios to verify the stability of the system.

**Author Contributions:** Formal analysis, C.P.; Project administration, H.L.; Resources, S.C.; Software, C.P.; Validation, C.P. and S.C.; Writing—original draft, C.P.; Writing—review and editing, C.P. and H.L. All authors have read and agreed to the published version of the manuscript.

**Funding:** This research received no external funding.

**Acknowledgments:** This work was supported by "The Technology Innovation Program" (10052501, Development of design technology of a device visualizing the virtual driving environment and synchronizing with the vehicle actual driving conditions to test and evaluate ADAS) funded by the Ministry of Trade, Industry, and Energy (MI, Korea).

**Conflicts of Interest:** The authors declare no conflict of interest.

## Appendix A

The ray casting algorithm is one of the most popular methods that distinguishes whether a point is inside or outside a polygon. It is also known as the crossing number algorithm or the even–odd

rule algorithm. It can be determined by infinitely drawing the ray from the point in any direction and counting the number of path segments the ray crossing. The point is outside if the counting number is even and inside if the number is odd. The example in Figure A1 shows the ray from the outside point $P_m$ has two crossing points on $\overline{A_3A_4}$ and $\overline{A_{14}A_{15}}$ as even numbers. On the other hand, the ray from $P_n$ intersects three path segments $\overline{A_5A_6}$, $\overline{A_6A_7}$ and $\overline{A_7A_8}$. It means that the point $P_n$ is inside the area.

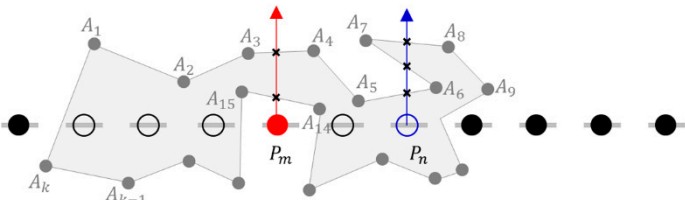

**Figure A1.** An example of the ray casting algorithm.

Another is the winding number algorithm. First, it needs to find all the path segments that crossed the ray from the point like the previous algorithm. Next, 1 is assigned to each path segment if its direction as viewed from the point is clockwise, and −1 is assigned where the direction is counterclockwise. Finally, the winding number is the accumulated sum of these. If the winding number is zero, the point is outside the polygon. Otherwise, the point exists inside. The winding number of $P_m$ is zero because $\overrightarrow{A_3A_4}$ is assigned −1 and $\overrightarrow{A_3A_4}$ is assigned 1. So, this point is outside the area. The winding number of $P_m$ is 1 as nonzero and it exists inside.

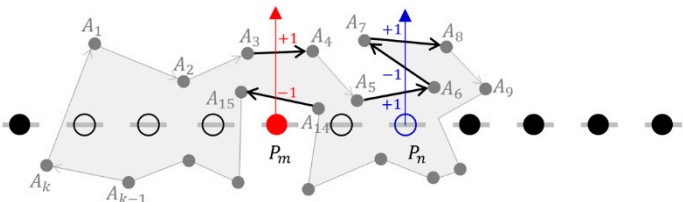

**Figure A2.** An example of the winding number algorithm.

**Appendix B**

The visualization module receives the pose data of all objects in the virtual and the actual test vehicle in the real-world. These data were used to visualize and were provided to the driver as a video. The information is shared based on Ethernet communication to benefit the programs running in a PC environment. In particular, fast data transmission through UDP and the scalability of simultaneously acquiring information from multiple devices through broadcast are expected. The packet loss of the UDP communication was neglected because this module needed only the latest data to update the virtual scene and the transfer rate was about 1.5 times more than the requirement of the visualization module. Applying the real-time pose information of the objects received from the virtual driving environment with a high frequency to the object can demonstrate the effect of continuous movement. At this time, there is a virtual camera that depends on the test vehicle objects, so it can create an image from the driver's perspective. This module allows the driver to recognize real-time information of the virtual driving environment through a display device, such as a monitor or HMD. Figure A3 shows the structure of the driver visualization module through Unreal Engine 4.

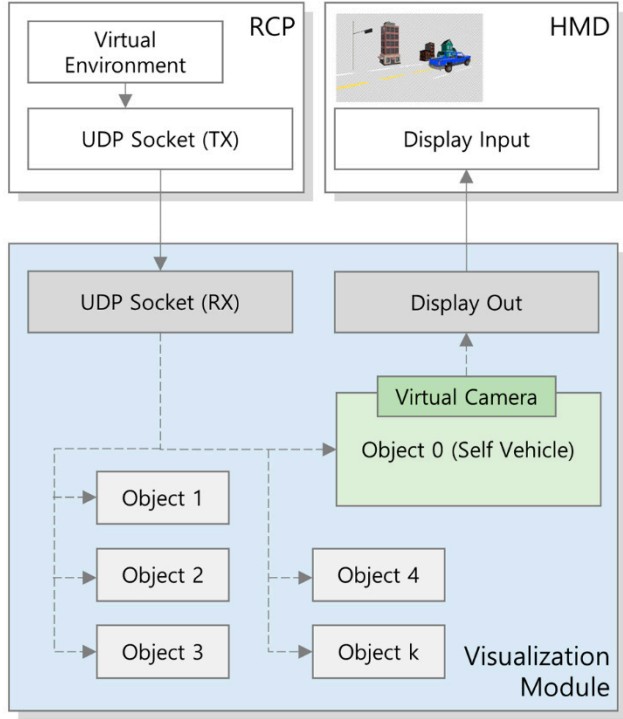

**Figure A3.** Visualization module structure.

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
