# Peer review of "Vehicle-in-the-Loop in Global Coordinates for Advanced Driver Assistance System"

_applsci, doi:10.3390/app10082645_

Round 1

Reviewer 1 Report

It would be a good idea to identify closely the equipment implemented in the testing phase of the research.

Maybe the phrase ”Interact with the vehicle's external environment.” should be completed in the abstract. 

The images in the figures 2, 3, 4 and 5 should have a higher resolution, and their text must be readable. 

Author Response

Response to Reviewer 1 Comments

First of all, we would like to thank the Reviewers for their high quality and constructive reviews of our manuscript, and the Editor for his careful reading. In this revised version of the manuscript, we did our best to address all comments raised by the Reviewers. A detailed item-by-item response to the Reviewers’ points follows. Also, the revised manuscript is uploaded attachment. The revisions are highlighted using the “Track Changes” function in Microsoft Word as the editors’ request.

Point 1: Maybe the phrase “Interact with the vehicle's external environment.” should be completed in the abstract.

Response 1: It is removed because It was repeated from the preceding sentence. (line 16, page 1)

Point 2: The images in the figures 2, 3, 4 and 5 should have a higher resolution, and their text must be readable.

Response 2: We have changed all figures with a higher resolution and made font size up to be readable.

Reviewer 2 Report

This paper presents the work on VIL for validating ADAS functions. This problem is important for developing vehicles with high levels of automation. Below are my comments.

When dealing with VIL problems, the fidelity of simulated sensor measurements is important for ensuring the reliability of perception in real environment. However, in this work, it seems that perfect sensor measurements are utilized without any noises or environmental impacts (e.g., light, shadow, occlusions). This will reduce the confidence about whether the VIL tests can imply the results in practice.

It seems that the VIL tests in this paper did not include responses of other vehicles to the ego vehicle. In practice, there are interactions with the ego vehicle and the surrounding road users. For simple tests such as AEB and lane keeping, it is fine to ignore the interactions in simulation. But in general, it is important to include interactions.

In the experiments, what is the update frequency for the simulation to provide environmental information to the ego vehicle? Moreover, I am interested in how large area the simulation environment can cover.

Figure A1 shows that the internal connection is through UDP. Then, how to deal with packet drops in UDP connection?

Author Response

Response to Reviewer 2 Comments

First of all, we would like to thank the Reviewers for their high quality and constructive reviews of our manuscript, and the Editor for his careful reading. In this revised version of the manuscript, we did our best to address all comments raised by the Reviewers. A detailed item-by-item response to the Reviewers’ points follows. Also, the revised manuscript is uploaded attachment. The revisions are highlighted using the “Track Changes” function in Microsoft Word as the editors’ request.

Point 1: When dealing with VIL problems, the fidelity of simulated sensor measurements is important for ensuring the reliability of perception in real environment. However, in this work, it seems that perfect sensor measurements are utilized without any noises or environmental impacts (e.g., light, shadow, occlusions). This will reduce the confidence about whether the VIL tests can imply the results in practice.

Response 1: We have added the noise term for the radar sensor and the real environmental impact to the vision sensor. We have shown how to implement it in our suggested system. Also, some models are introduced for advanced systems. (line 194~, page 7) (line 215~, page 8)

Point 2: It seems that the VIL tests in this paper did not include responses of other vehicles to the ego vehicle. In practice, there are interactions with the ego vehicle and the surrounding road users. For simple tests such as AEB and lane keeping, it is fine to ignore the interactions in simulation. But in general, it is important to include interactions.

Response 2: We have added the simple vehicle model for the responses of vehicles. Also, Gipps’s model is introduced for the driver model to generate input of the vehicle model. (line 143~, page 5)

Point 3: In the experiments, what is the update frequency for the simulation to provide environmental information to the ego vehicle? Moreover, I am interested in how large area the simulation environment can cover.

Response 3: We have added the update times, resolutions and coverages of every module. The coverages of the area can be as much the engineer needs because it depends on data size. (line 255, page 9)

Point 4: Figure A1 shows that the internal connection is through UDP. Then, how to deal with packet drops in UDP connection?

Response 3: We have added explain about UDP pack drops. We neglect the problem because the system uses only the latest data and any lost data does not affect the current state. (line 331, page 12)

Round 2

Reviewer 2 Report

The authors have addressed the reviewer's comments.